# Evaluating Road Lighting Quality Using High-Resolution JL1-3B Nighttime Light Remote Sensing Data: A Case Study in Nanjing, China

**Nuo Xu, Yongming Xu \*, Yifei Yan, Zixuan Guo, Baizhi Wang and Xiang Zhou**

School of Remote Sensing and Geomatics Engineering, Nanjing University of Information Science and Technology, Nanjing 210044, China
* Correspondence: xym30@nuist.edu.cn

**Abstract:** A good lighting environment for roads at night is essential for traffic safety. Accurate and timely knowledge of road lighting quality is meaningful for the planning and management of urban road lighting systems. Traditional field observations and mobile observations have limitations for road lightning quality evaluation at a large scale. This study explored the potential of 0.92 m resolution JL1-3B nighttime light remote sensing images to evaluate road lighting quality in Nanjing, China. Combined with synchronous field measurements and JL1-3B data, multiple regression and random forest regression with several independent variable combinations were developed and compared to determine the optimal model for surface illuminance estimation. Cross validation results showed that the random forest model with Hue, saturability, ln(Intensity), ln(Red), ln(Green) and ln(Blue) as the input independent variables had the best performance ($R^2$ = 0.75 and RMSE = 9.79 lux). Then, this model was used to map the surface illuminance. The spatial scopes of roads were extracted from Google Earth images, and the illuminance within roads was derived to calculate the average, standard deviation and coefficient of variation to indicate the overall brightness level and brightness uniformity of the roads. This study provides a quantitative and effective reference for road lighting evaluation.

**Keywords:** nighttime light remote sensing; road lighting quality measurements; JL1-3B; Nanjing

## 1. Introduction

In night driving, road night lighting plays an important role in traffic safety. The International Commission on Illumination (CIE) reported that road lighting reduced nighttime accidents by 13~75% over 15 countries [1]. Christopher found that the reported crashes where roads were lit decreased 28.95% in total crashes and 39.21% in injury night crashes [2]. Elvik et al. indicated that road lighting reduced the nighttime crash rate by 23% in Belgium, Britain and Sweden [3]. William found that improving overall uniformity up to approximately 0.4 lowers the night-to-day crash ratio for highways in New Zealand [4]. Good road lighting quality provides a good lit environment, thus reducing traffic crashes at night. Consequently, evaluating road lighting quality is meaningful to improve the nighttime road lit environment and gain urban traffic safety [5–9].

Road lighting quality reflects the photometric performance of road lights aiming at satisfying drivers' visual needs at night, which includes parameters such as average illuminance, overall uniformity, etc., according to CIE [10] and European Committee for Standardization [11] documents.

Field observation is the most popular way to evaluate road lighting quality. Generally, illuminance meters or imaging photometers are employed to measure road brightness at typical places to assess lighting quality. Liu et al. utilized different illuminance meters and handheld luminance meters to measure the illuminance in different orientations and the brightness of roads in Dalian, China [12]. Guo et al. measured the luminance of roads in Espoo, Finland using an imaging photometer LMK Mobile Advanced (IPLMA),

which converted photos into luminance values, and calculated the average luminance by the software LMK 2000 [13]. Jägerbrand also applied the IPLMA to obtain road lighting parameters in Sweden [14]. Ekrias et al. used an imaging luminance photometer ProMetric 1400 to measure the luminance of roads in Finland and calculated the average luminance and uniformities [15]. However, static road lighting measurements cannot provide lighting information at large scales. Additionally, observation positions must be placed within roads, which may affect the normal running of traffic flow [16–18].

Some researchers have carried out mobile measurements with imaging luminance devices or photometers boarded on vehicles. Greffier et al. used the High Dynamic Range (HDR) Imaging Luminance Measuring Device (ILMD) mounted on a car to measure road luminance [19]. Zhou et al. developed a mobile road lighting measurement system with a photometer mounted on a vehicle, which was able to record the illuminance and position data simultaneously and successfully applied in Florida, America [20]. However, mobile measurements are disturbed by many factors, such as the head and rear lights of cars, relative positions between observation vehicles and street lamps, and the vibration of observation angles.

Nighttime lighting remote sensing offers a unique way to monitor spatial-continuous nighttime nocturnal lighting at a large scale. Nighttime light remote sensing data, such as DMSP/OLS, NPP/VIIRS and Luojia1-01, are widely used in mapping urbanization processes [21], estimating GDP, investigating poverty and monitoring disasters [22]. However, due to the coarse resolutions of these nighttime satellite remote sensing data (DSMP/OLS: 2.7 km; NPP/VIIRS: 0.75 km; Luojia1-01: 130 m), they cannot map the detailed light environment within roads and therefore cannot be applied for road lighting quality evaluation, thus few studies have been carried out on road lighting environments based on nighttime light data However, there still are some studies. Cheng et al. used JL1-3B nighttime light data to extract and classify the street lights by a local maximum algorithm, achieving an accuracy of above 89% [23]. Zheng et al. used the multispectral feature of JL1-3B data to discriminate light source types using ISODATA algorithm, with an overall accuracy of 83.86% [24]. An unmanned aerial vehicle (UAV) is able to provide high-resolution nighttime lighting images on a relatively large scale. Rabaza et al. used a digital camera onboard the UAV to capture orthoimages of the lit roads in Deifontes, Spain, which was calibrated by a known luminance relationship. Then, the average luminance or illuminance was calculated [25]. UAVs have the outstanding advantages of convenience, low cost and extremely high resolution. However, it is also limited by security, privacy factors, and the short flight distance makes it unsuitable for large areas. The newly launched JL1-3B satellite provides nighttime images with a high spatial resolution of 0.92 m and multispectral data [24,26]. The features of JL1-3B show the potential of evaluating road lighting quality at large scales.

This study aimed to evaluate the road lighting quality in Nanjing, China, using high-resolution JL1-3B nighttime light remote sensing data combined with the in situ measured illuminance on typical roads. Several machine learning models were developed and compared to produce a fine resolution illuminance map with good accuracy. Then, the remotely sensed illuminance within roads was extracted to calculate indicators to evaluate road lighting quality.

## 2. Study Area and Datasets

### 2.1. Study Area

Nanjing, the capital city of Jiangsu Province, China, is located in eastern China. It covers a total area of 6587.02 km$^2$, with latitudes ranging from 31°14′ to 32°37′N and longitudes ranging from 118°22′ to 119°14′E. In recent decades, Nanjing has experienced significant economic and population growth and, accordingly, a rapid increase in the number of motor vehicles. In 2020, the total number of motor vehicles reached 2,799,469, and the total length of roads reached 9796 km [27]. It is difficult to evaluate the lighting quality of such a vast road network by timely in situ observations. Therefore, exploring

an effective method for road lighting assessment at a large scale is very meaningful for Nanjing city.

### 2.2. Data Collection

#### 2.2.1. Remote Sensing Data

JL1-3B, known as "Jilin-1 03B", which was developed by Changguang Satellite Technology Co., Ltd. (Changchun, China) and launched on 9 January 2017. This satellite is on a sun-synchronous orbit with an altitude of 535 km, which provides three imaging models: video imaging, push-broom imaging and night light imaging [23]. Night light imaging models provide radiometrically and geometrically calibrated high spatial resolution (0.92 m) images with multiple bands (red band: 580–723 nm, green band: 489–585 nm, blue band: 437–512 nm). Regions within an off-nadir angle of $\pm 45°$ can be accessed by the area array camera boarded on the payload. Each tile of the image covers 11 km $\times$ 4.5 km of the ground area. Compared with DMSP-OPS, NPP-VIIRS and LJ1-01 nighttime light remote sensing satellites, the high spatial resolution of JL1-3B is capable of depicting the spatially detailed lit environment within road lights. Compared with the high-resolution nighttime light remote sensing satellite EROS-B (0.7 m), multispectral data are provided, allowing for more detailed studies relative to road lighting [23,24,26,28]. The JL1-3B data can be commercially ordered from the website of Changguang Satellite Technology Co., Ltd. (http://www.jl1.cn/Search.aspx?txtSearch=JL1-3B (accessed on 5 October 2020)), of which price is 180 CNY/km$^2$.

The JL1-3B image in this study was collected at 21:08:37 (Beijing Time) on 23 October 2020, which covered the northern part of the main urban area (Figure 1). The off-nadir angle was 0.02 degrees, avoiding the shielding effect of buildings on the road lit environment. The image was primarily geometrically and radiometrically corrected (Figure 2).

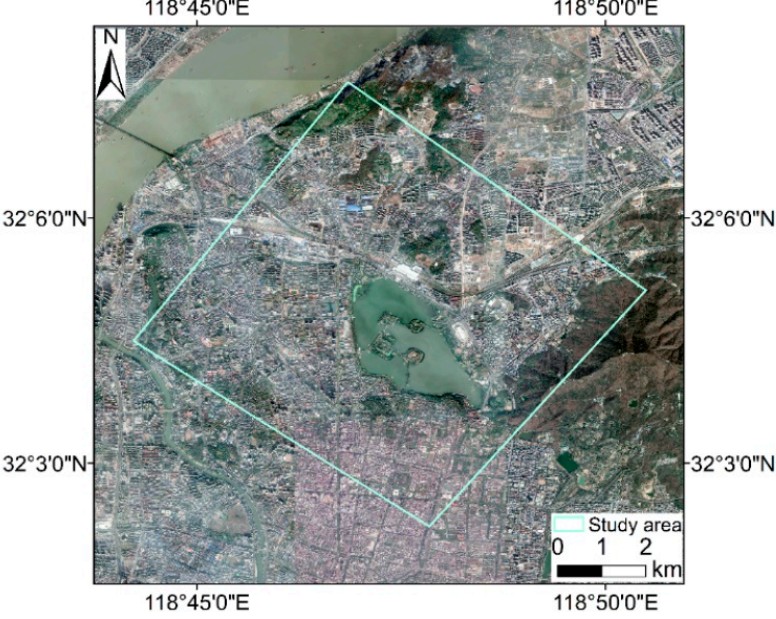

**Figure 1.** JL1-3B image range of study area without geometric correction (base map: Google Earth).

#### 2.2.2. Field Measurements

Field observations were carried out from 20:30–21:30 on 23 October 2020, to coincide with the JL1-3B overpass time. A TES-1399R illuminance meter was employed to measure the in situ illuminance. Its measurement range is from 0.01 lux to 999,900 lux, and the accuracy is $\pm 3\%$ rdg (calibrated to standard incandescent lamp, 2856K). Five groups carried out observations on the foot with TES-1399R simultaneously on different representative routes, which covered different lighting conditions. At each sample point, the TES-1399R

was facing-upward horizontally, placed at a height of 1.5 m to measure the downward illuminance from the street lamps. The locations of these sample points were also recorded by Global Navigation Satellite System (GNSS) instruments. To avoid the influence of the shading effect of street trees, only the sites that were not covered by trees were selected for observation. A total of 214 measurements were collected (Figure 2).

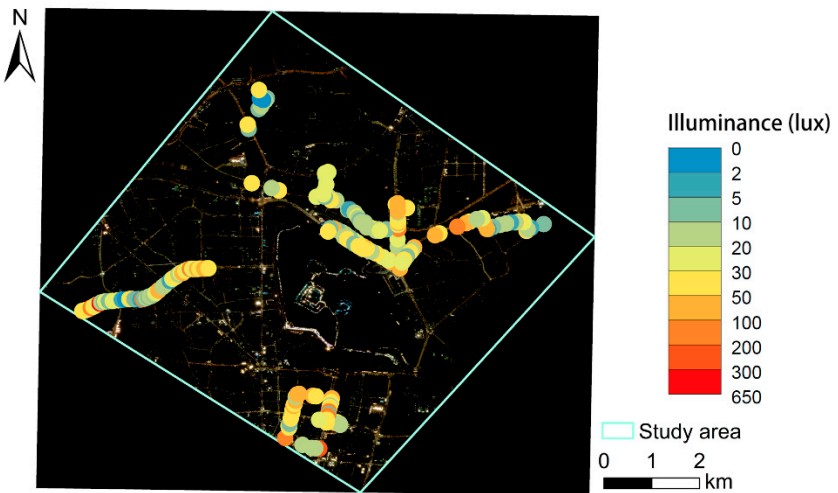

**Figure 2.** Distribution of measured points (base map: raw JL1-3B image).

## 3. Methodology

### 3.1. Workflow

The flowchart of this study is shown in Figure 3, which includes three main steps: (1) preprocessing the JL1-3B data; (2) developing multiple models and selecting the best of them to map illuminance from the JL1-3B image; and (3) evaluating road lighting quality based on the illuminance map.

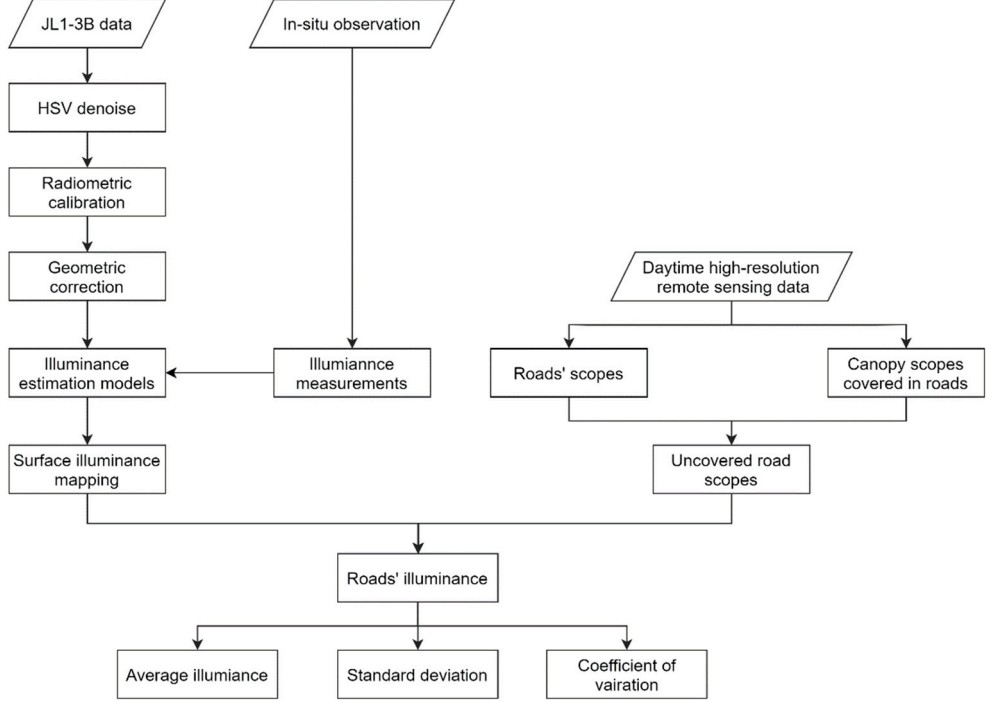

**Figure 3.** Flow chart.

### 3.2. JL1-3B Data Preprocessing

The noise was removed using the threshold method in the HSV color space. Noises consist of a patch of pixels with high values of the R, G or B band, which randomly cross the image. Considering that they are characterized by high saturation, the image was transformed from RGB to HSV space to identify noisy pixels by the following rules:

$$\{|H-120|<10 \ or \ |H-240|<10 \ or \ |H-360|<10\} \ and \ \{S>0.8\} \ and \ \{V>0.55\}, \quad (1)$$

where $H$ is the hue, $S$ is the saturation and $V$ is the value of the HSV space.

The noisy pixels were identified and then filled using Delaunay triangulation with triangles calculated from the surrounding valid DN values. Figure 4b shows the denoised image. After the denoising process, the monochromatic noisy pixels were effectively removed. Note that the denoising process should be performed before geometric registration because resampling during the registration will mix the monochromatic noisy values and normal values, making it difficult to distinguish noisy pixels.

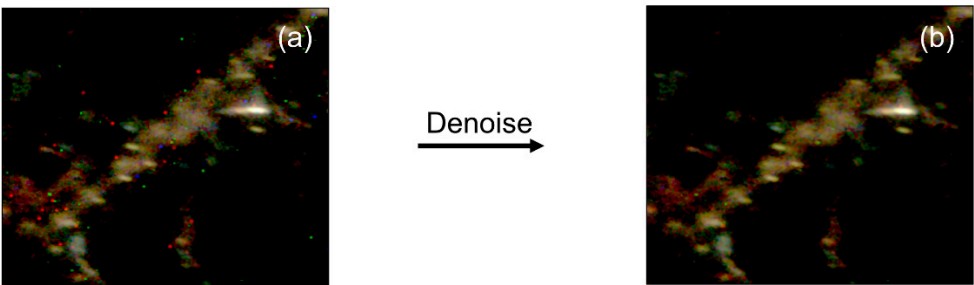

**Figure 4.** Comparison before (**a**) and after denoising (**b**).

The denoised JL1-3B image was calibrated in radiation according to Equation (2):

$$L_i = \frac{D_i - b_i}{a_i}, \quad (2)$$

where $L_i$ is the radiance (W·m$^{-2}$·sr$^{-1}$) of band $i$, $D_i$ is the DN value of band $i$, and $a_i$ and $b_i$ are the calibrated coefficients of band $i$ (Table 1).

**Table 1.** Radiance calibration coefficients of JL1-3B.

| Wavelength | a | b |
|---|---|---|
| Band 1 (Red) | 9681 | −4.73 |
| Band 2 (Green) | 5455 | −3.703 |
| Band 3 (Blue) | 2997 | −4.471 |

Although the JL1-3B L1A image has been systematically geometrically corrected, it still has obvious geometric deviations. Taking a high-resolution Google Earth image as the reference image, 26 ground control points were selected to register the JL1-3B image. It should be noted that the JL1-3B sensor is an array imaging sensor, and the JL1-3B image of the study area was mosaicked from multiple images. Therefore, the geometric distortion of the whole image varies from different source images. Under this consideration, the Delaunay triangulation method instead of the polynomial method was employed for image wrapping. Figure 5 shows the registered JL1-3B image.

During field observations, the surrounding environment is complex, and the measured values may be affected by a variety of factors, such as the headlights of passing vehicles, landscape lighting and window lights from nearby buildings. These interference factor may affect the consistency between surface observed illuminance and space-born observed illuminance. For example, the in-situ observations that were illuminated by vehicle headlights had relative high measurement values, but the corresponding pixel values in the

JL1-3B image were not high because the filed observation and the satellite overpass were not perfectly synchronous. Therefore, all the sample points were manually checked to remove problematic sample points to improve the estimation accuracy of illuminance. Finally, 168 sample points remained.

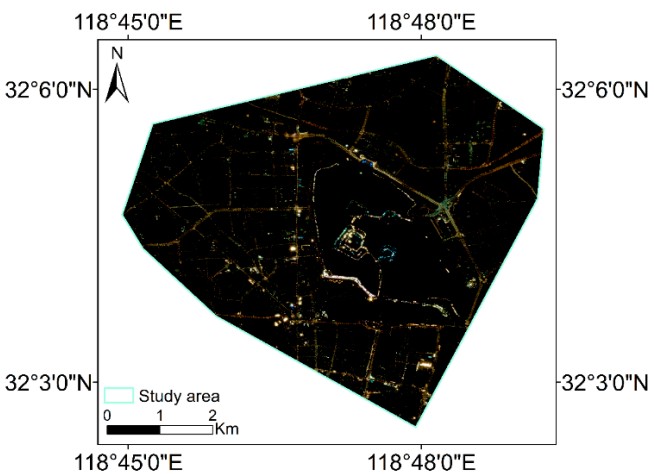

**Figure 5.** The JL1-3B image after preprocessing.

### 3.3. Illuminance Mapping

The emitted light from street lamps is reflected by the road surface and then propagates through the atmosphere to reach the satellite sensor. The road surface has similar reflection characteristics, and the atmospheric conditions are relatively uniform at a small scale. In this way, a close relationship between the in situ observed illuminance and the remotely sensed radiance should exist. Figure 6a gives the scatter plots between the observed illuminance and the radiance of the three JL1-3B bands. The illuminance has good relationships with the radiance of the R, G and B bands, with correlation coefficients of 0.638, 0.648, and 0.649, respectively. It can also be noted that there are many unique values in band B, which can be attributed to the relatively strong scattering intensity of the atmosphere for the blue band [29,30]. Figure 6b gives the scatter plots between the observed illuminance and the log values of the radiance of the three bands. The correlations between them were higher than those between the observed illuminance and the radiance values. In order to provide more relevant variables to the illuminance estimation model, HIS color space was introduced under the consideration that it is more robust to changes in illuminance [31]. Thus, the RGB color space was transferred to HIS color space, and the correlation coefficients between illuminance and H, I, S were −0.34, 0.67 and −0.37, respectively. After their logarithmic transformation, log(I) showed the highest correlation coefficient (0.77) with illuminance, and the Pearson coefficients of log(H) and log(S) were both −0.34.

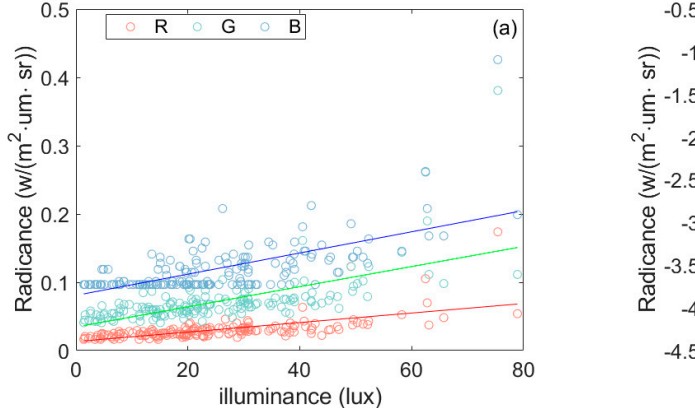
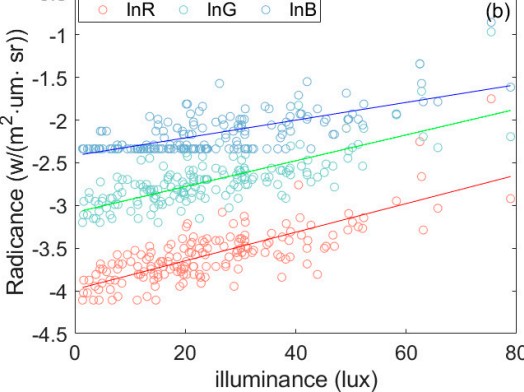

**Figure 6.** (**a**) RGB scatter plot; and (**b**) lnRGB scatter plot.

Relationships between illuminance and different color components varies [32]. Based on the correlation analysis above, R, G, B, I and their log values show high correlations with observed illuminance. Considering the auxiliary information that the variable Hue and Saturation can also provide, variables in RGB color space and HIS color space were integrated. Then, six combinations of these independent variables were employed (Table 2) to determine the optimal variables for illuminance estimation.

**Table 2.** Combinations of independent variables.

| Combination | Independent Variables |
| --- | --- |
| 1 | R, G, B |
| 2 | lnR, lnG, lnB |
| 3 | H, S, I |
| 4 | H, S, lnI |
| 5 | H, S, I, R, G, B |
| 6 | H, S, lnI, lnR, lnG, lnB |

To properly portray the quantitative relationships between variables in Table 2 and illuminance, multiple linear regression (MLR) and random forest (RF) algorithms were used to develop models for illuminance mapping based on the abovementioned variable combinations. MLR is widely used for its simplicity and interpretability [33,34]. Random forest regression (RF) is an ensemble algorithm combined with tree predictors such that random inputs and features are selected in the process of forming trees [35]. The predicted result is the average value of the overall regression trees, which can effectively reduce the bias of a single tree [36].

Taking the observed illuminance as the dependent variable and the six variable combinations as the independent variables, multiple linear regression and RF models were fitted. The predictive accuracy was evaluated using 10-fold cross-validation. First, the dataset was randomly divided into 10 uniform subsets. Then, one subset (17 samples) was used as the test set, and the remaining 9 subsets (151 samples) were used as the training set to fit the model. This process was repeated 10 times using each subset as the test set once. *RMSE* (Equation (3)) and $R^2$ (Equation (4)) were calculated to indicate the model performance.

$$RMSE = \sqrt{\frac{\sum_{\{i=1\}}^{n}(\hat{y}_i - y_i)^2}{n}},\qquad(3)$$

$$R^2 = 1 - \frac{\sum_{i=1}^{n}(\hat{y}_i - y_i)^2}{\sum_{i=1}^{n}(y_i - \overline{y})^2},\qquad(4)$$

where $\hat{y}_i$ is the predicted illuminance (lux), $y_i$ is the actual observed illuminance (lux), $\overline{y}$ is the average illuminance (lux) of the whole road, and $n$ is the number of samples.

The number of weak classifiers ($n$) and the number of variables selected randomly for each tree split ($s$) are two important parameters of RF. The parameter $n$ influences the fitting effectiveness [37], and the parameter s affects the final results, both of which were tuned to optimize the model

By comparing the accuracies of the models, the best model and variable combination were chosen. Then, this model was applied to the corresponding JL1-3B spatial variables to produce a high-resolution surface illuminance map of the study area.

### 3.4. Road Lighting Quality Evaluation

The spatial scopes of the different classes of roads where distribution of illuminance was obvious and continuous were extracted from Google Earth images by visual interpretation. Note that different sections of one road may vary in different classes and lighting conditions; thus, sections of roads were considered as the evaluating objects. During the process of extracting roads, some rules were obeyed. (1) The intersection areas with buffers

of approximately 20 m were not extracted to avoid the influence of mixed-source lighting; (2) the street tree canopies were also excluded in order to avoid the shaded pixels within the roads (Figure 7); and (3) due to the relatively low accuracy of nighttime lighting data geometrical correction, roads with relatively low degrees of overlap between the JL1-3B preprocessed image and Google Earth image (base map in the preprocess of geometric correction) were excluded.

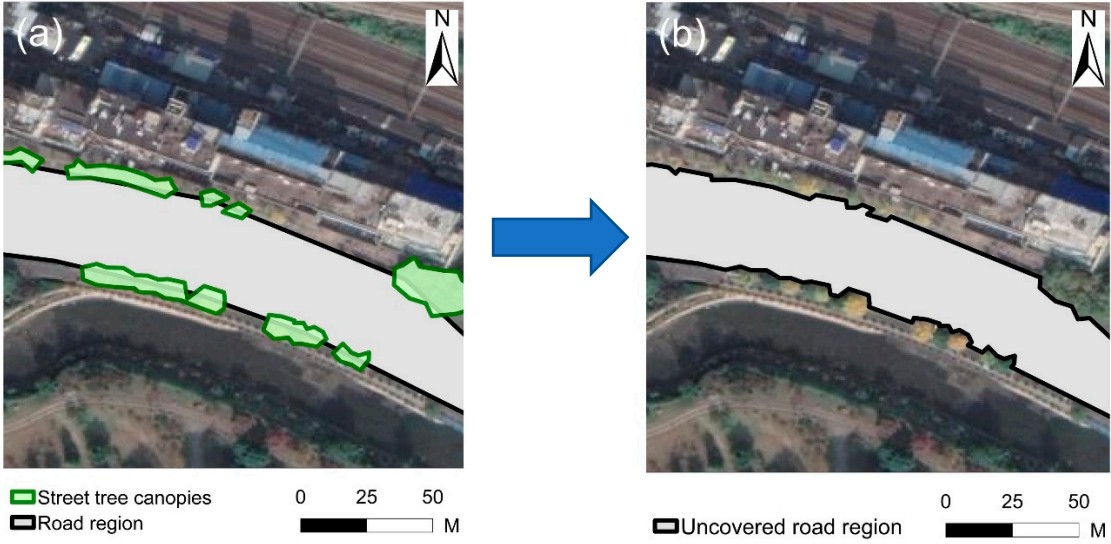

**Figure 7.** Extraction process illustration of uncovered road region: (**a**) extracted road region and street tree canopies; (**b**) uncovered road region.

Based on the remotely sensed illuminance map and road scopes, the mean value, standard deviation and coefficient of variation ($C_v$) of the illuminance within each road were calculated. The average illuminance indicates the overall brightness level of the road, and the standard deviation and $C_v$ reflect the brightness uniformity of the road.

$$\overline{E} = \frac{\sum_{i=1}^{n} E_i}{n},$$ (5)

$$\sigma = \sqrt{\frac{\sum_{i=1}^{n} \left(E_i - \overline{E}\right)^2}{n}},$$ (6)

$$C_v = \frac{\sigma}{\overline{E}},$$ (7)

where $\overline{E}$ is the average illuminance of the road or road section (lux), $E_i$ is the illuminance of pixel $i$ in the road or road section (lux), and $n$ is the number of pixels in the road or road section. $\sigma$ is the standard deviation of the illuminance of the road or road section (lux), and $C_v$ is the coefficient of variation of the illuminance of the road or road section.

The standard for lightning design of urban roads in China [38] specifies the standards of overall brightness and uniformity. The overall brightness is identified by average illuminance and the threshold values of different classes of road were shown in Table 3. For expressways and main roads, the average illuminance is required to be greater than 20 lux. For secondary roads, the average illuminance should be more than 10 lux, and for branches, the average illuminance should be greater than 8 lux. The uniformity is identified by the ratio of the minimum illuminance to average illuminance. However, this factor is not suitable for remote sensing. The minimum value is not stable, which is easily affected by image noise, shades and other factors. Additionally, it cannot fully utilize the advantages of remote sensing that can map spatial continuous lit environments. In this study, we

employed standard deviation and $C_v$ of illuminance to indicate uniformity, and took the mean values of $C_v$ as the reference values for road lighting uniformity.

**Table 3.** Average illuminance standard for road grades [18,38].

| Road Grade | Average Illuminance Standard |
|---|---|
| Expressway/Main road | ≥20 lux |
| Secondary road | ≥10 lux |
| Branch | ≥8 lux |

## 4. Results

### 4.1. Surface Illuminance Map

Cross-validation results with goodness of fit ($R^2$) and root mean squared error (RMSE) of MLR and RF models based on different variable combinations are shown in Table 4, and the optimal parameters of RF for each variable combination are listed in Table 5. In terms of $R^2$, the performance of the RF models was generally better than that of MLR, for which $R^2$ for all variable combinations was 0.74 or 0.75. Especially for variable combination 1, the $R^2$ rose from 0.47 (MLR model) to 0.74 (RF model), indicating that the nonlinear relationship between illuminance and the R, G, and B values was more appropriate than the linear relationship. With regard to the performance of MLR models, variable combination 6 was the optimal with the highest $R^2$ (0.64) and lowest RMSE (9.50 lux), and variable combination 5 was slightly worse than it. From the perspective of variable combinations' color spaces, variable combinations 5 and 6 outperformed variable combinations 3 and 4 as well as combinations 1 and 2 in both the MLR and RF models, suggesting that the models combining RGB and HIS color spaces were superior to the single-color space. Based on the above analysis, RF models with variable combinations 5 and 6 were relatively better choices for estimating the illuminance. Considering that the RMSE of variable combination 5 was only 0.03 smaller than that of variable combination 6, and this study preferred $R^2$ as the more important metric for models, variable combination 6 applying the RF model with the highest $R^2$ (0.75) and the relatively low RMSE (9.79 lux) was selected as the optimal model for further estimation.

**Table 4.** Validation results of the models with different variable combinations.

| Models | | MLR | | RF | |
|---|---|---|---|---|---|
| Number | Combinations | $R^2$ | RMSE | $R^2$ | RMSE |
| 1 | R, G, B | 0.47 | 11.36 | 0.74 | 9.72 |
| 2 | lnR, lnG, lnB | 0.62 | 9.55 | 0.74 | 9.72 |
| 3 | H, S, I | 0.50 | 10.98 | 0.75 | 10.03 |
| 4 | H, S, lnI | 0.62 | 9.62 | 0.75 | 10.03 |
| 5 | H, S, I, R, G, B | 0.63 | 9.52 | 0.74 | 9.76 |
| 6 | H, S, lnI, lnR, lnG, lnB | 0.64 | 9.50 | 0.75 | 9.79 |

**Table 5.** Optimal parameters for RF models with different variable combinations.

| Combination No. | Number of Classifiers (n) | Number of Variables (s) |
|---|---|---|
| 1 | 150 | 3 |
| 2 | 150 | 3 |
| 3 | 110 | 3 |
| 4 | 110 | 3 |
| 5 | 110 | 6 |
| 6 | 120 | 6 |

Figure 8 shows the scatter plot between the observed and estimated illuminance from the random forest model based on variable combination 6. Most samples were clustered

near the 1:1 line, indicating the good fitness of the model. In addition, there was no obvious overestimation or underestimation.

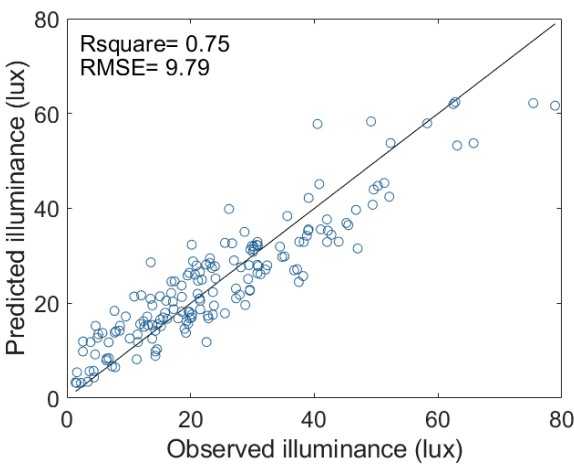

**Figure 8.** Scatter plot for the optimal model.

The developed random forest model with variable combination 6 was applied to the corresponding spatial independent variables, and the surface illuminance over the study area was mapped (Figure 9).

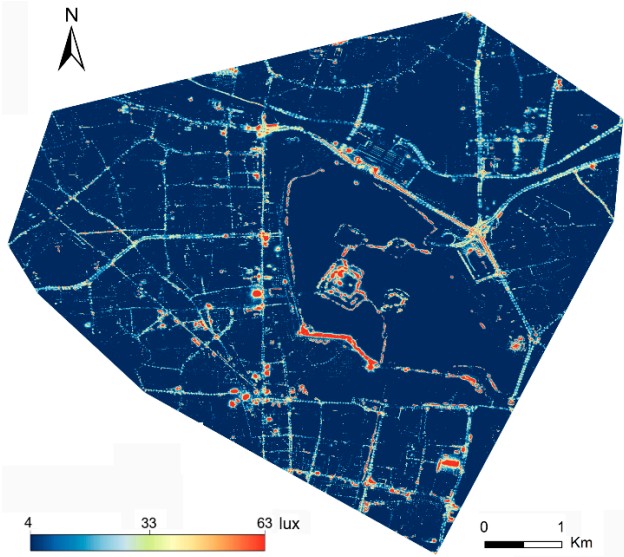

**Figure 9.** Illuminance map.

### 4.2. Road Lighting Quality Evolutions

The main 50 roads in the study area were extracted manually, including 16 expressways/main roads, 12 secondary roads and 22 branches. Based on the remotely sensed surface illuminance map and road spatial scopes, the illuminance values within each road were extracted to calculate the average, standard deviation and $C_v$ to assess lighting quality. The results are shown in Table S1. Additionally, the boxplots of the illuminance within each road are shown in Figure S1. Table 6 shows the statistical results of the lighting quality the expressways/main roads, secondary roads and branches. The mean of average illuminance of expressways/main roads, secondary roads and branches were 20.64 lux, 22.35 lux, and 19.46 lux, respectively. Generally, the overall average illuminance of all the three road classes met the average illuminance standard shown in Table 3. However, the minimum values of average illuminance of expressways/main roads, secondary roads and branches were 9.90 lux (Heyan Road), 12.08 lux (Hunan Road), and 8.58 lux (Gaomenlou

Road), respectively, indicating that there were some expressways/main roads and secondary roads that did not meet the average illuminance standard. Overall, 87.5% of the expressways/main roads, all the secondary roads and all the branches met the standard of average illuminance.

**Table 6.** The statistics of road lighting quality indicators for each road grade.

| Grade | Road Lighting Quality Indicators | Maximum | Minimum | Mean |
|---|---|---|---|---|
| **Expressway/Main road** | Average illuminance(lux) | 27.74 | 9.90 | 20.64 |
| | Std (lux) | 14.12 | 8.28 | 10.89 |
| | $C_v$ | 0.98 | 0.42 | 0.56 |
| **Secondary road** | Average illuminance (lux) | 36.75 | 12.08 | 22.35 |
| | Std (lux) | 13.83 | 8.29 | 10.65 |
| | $C_v$ | 0.90 | 0.33 | 0.52 |
| **Branch** | Average illuminance (lux) | 36.13 | 8.58 | 19.46 |
| | Std (lux) | 12.79 | 5.30 | 9.51 |
| | $C_v$ | 0.89 | 0.28 | 0.54 |

From the aspects of standard deviation, which represents the absolute uniformity, the mean value of standard deviation of expressways/main roads, secondary roads and branches were 10.89 lux, 10.65 lux, and 9.51 lux, respectively. Though the three road classes showed similar overall standard deviation, their $C_v$ values were quite different because their average illuminances are different. In fact, absolute uniformity (standard deviation) was not suitable for the indicator that was used to compare illuminance variation in roads as it ignores the impact of average illuminance, thus, $C_v$ was considered as the better indicator to evaluate the uniformity of roads. The mean values of $C_v$ of expressways/main roads, secondary roads and branches were 0.56, 0.52 and 0.54, respectively. However, the maximum $C_v$ values of these three road classes were 0.98 (Heyan Road), 0.90 (Hunan Road) and 0.89 (Gaomenlou Road), suggesting that there were some roads that suffered poor uniformity of illuminance. Given that the existing standard road lighting uniformity is not proper for remotely sensed illuminance, we set $C_v$ of the three road classes (0.56, 0.52, and 0.54, respectively) as the reference values of lighting uniformity. The roads with $C_v$ higher than the reference values were considered that had poor road lighting uniformity. Overall, 75% of the expressways/main roads, 66.7% of the secondary roads and 50% of the branches showed good uniformity.

Four typical roads were selected for detailed analysis, including Jianning Road, Hubei Road, Hongshan Road and Longpan Road (Table 7). Jianning Road showed both poor overall brightness and uniformity; Hubei Road showed good overall brightness, but poor uniformity; Hongshan Road had poor overall brightness, but good uniformity; Longpan Road had both good overall brightness and uniformity.

**Table 7.** Values of typical roads' lighting quality evaluation metrics.

| Road Name | Average Illuminance (lux) | Std (lux) | $C_v$ | Grade |
|---|---|---|---|---|
| Jianning Road | 16.65 | 9.83 | 0.59 | Main road |
| Hubei Road | 23.03 | 12.79 | 0.56 | Branch |
| Hongshan Road | 19.27 | 8.51 | 0.44 | Expressway |
| Longpan Road | 27.74 | 13.44 | 0.48 | Expressway |

Figure 10 shows the illuminance maps (local segments) and histograms (all the roads) of these four roads, which presents more details at pixel scale than the table and boxplots. For Jianning Road (Figure 10a), the illuminance map shows that most pixels within the road had low illuminance values indicating the poor overall brightness, and unilluminated sections of road were interspersed, resulting in the awful uniformity. The histogram of this road illustrates that most pixels were under the standard for the main roads (20 lux), and the distribution was obviously skewed to a low illuminance value. For Hubei Road

(Figure 10b), most areas within the road were obviously de-lighted. However, the middle sections were much brighter than the sides. The histogram indicates that most pixels ranged from 15 to 35 lux and there were not relatively fewer extreme illuminated pixels. For Hongshan Road (Figure 10c), the road regions were approximately evenly and fully illuminated, but the overall brightness was relatively low. Most pixels were concentrated between 10–20 lux from the histogram of this road, and few pixels had extreme illuminance values. The illuminance map for Longpan Road (Figure 10d) showed a generally high and even distributed lit environment, and the histogram also indicated this point. The illuminance map and histogram can not only evaluate the lighting quality of the whole road, but can reflect more spatial and distribution details.

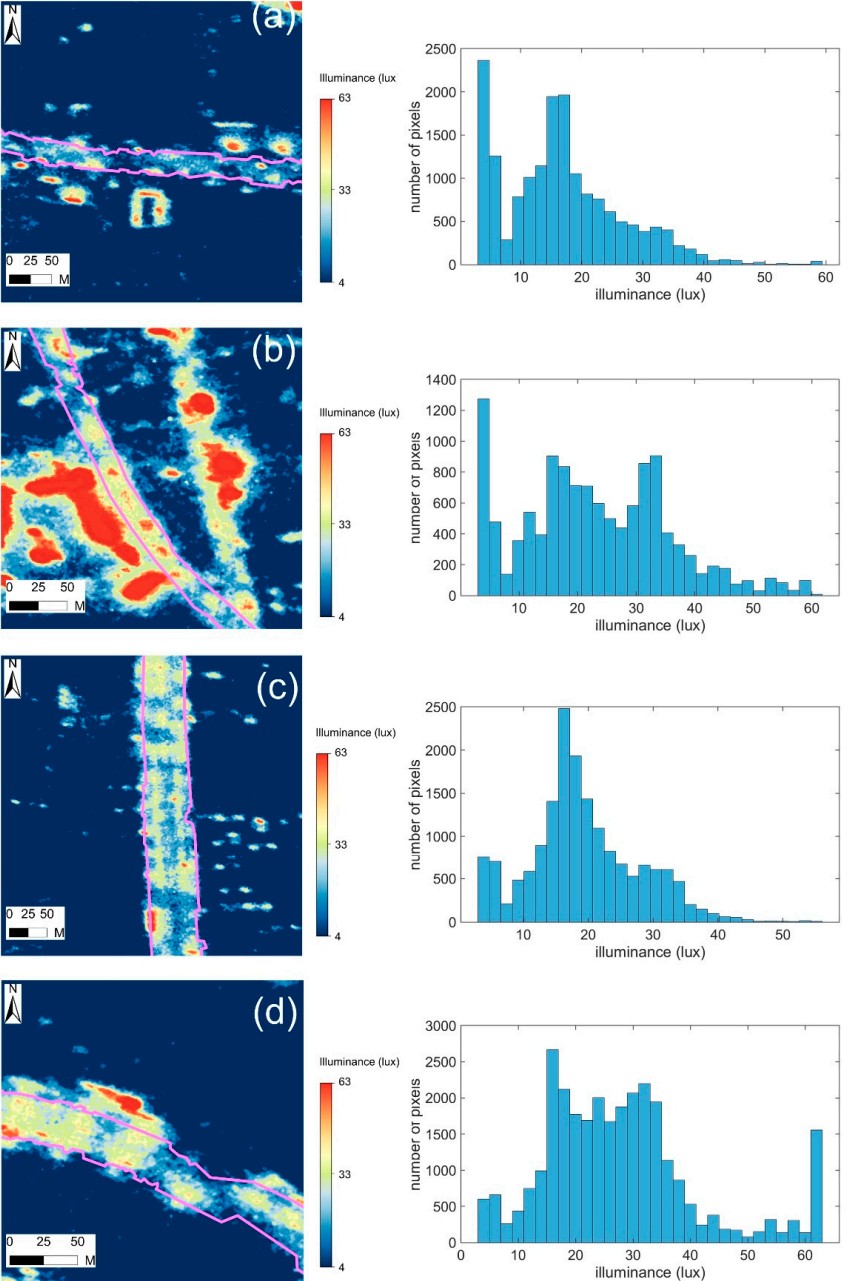

**Figure 10.** The illuminance maps and histograms of 4 typical roads. The purple box shows the roads' scopes: (**a**) Jianning Road; (**b**) Hubei Road; (**c**) Hongshan Road; and (**d**) Longpan Road. To illustrate the illuminance more clearly, illuminance maps show local sections of the road. Histograms represent the illuminance of the whole road.

## 5. Discussion

Traditional ground-based measurements are limited by the measuring scopes and usually disturb the normal traffic flow. Nighttime remote sensing can observe spatial continuous lit environments at large scales, providing potential for road lighting quality evaluation. However, due to the relative coarse resolutions, most nighttime remote sensing data cannot be used to map illuminance within roads. The new JL1-3B data has a high spatial resolution of 0.92 m that can capture detailed lighting environment within roads, and it also has three channels that can well characterize the light colors [23,24]. Therefore, JL1-3B is a proper data that can be used for effectively quantifying road lighting quality. This paper explores the method to evaluating road lighting quality by the estimated illuminance derived from JL1-3B data and in situ observations. The high-resolution illuminance map contains large amounts of illuminance data and can effectively depict the lighting environment within roads at pixel scale. Furthermore, with the revisiting period of 9 days, the proposed method using JL1-3B data is able to measure the road lighting quality periodically, providing timely information about large-scale road lighting condition for concerned government departments. Compared with traditional road lighting quality measurements, the application of nighttime light remote sensing data in this study is superior in terms of safety, rapidity, measuring scopes, measuring frequency and information content. In addition to the radiance values of the three bands of JL1-3B, other color components, such as HIS, were also introduced for estimating illuminance. Six combinations of variables were compared to determine the optimal variables combination. The results (Table 4) showed the optimal model involved in all color components, indicating that considering more color components may improve the performance of the estimation model. Furthermore, the random forest model outperformed the multiple regression model. This may be attributed to the fact that different street lamps emit light of different colors, and the tree-based random forest method that can handle different conditions under different branches is more suitable for this complex scenario.

There were also some limitations in this study. During the process of data collection, some deficiencies like discrepancies between the ground and satellite observations and low positioning accuracy of the GNSS system, have negative impacts on the outcomes of the methods. Satellite imaging was almost instantaneous, but in situ observation lasted approximately 1 h. During observation, the lit environment may change, resulting in inconsistencies between field observations and remote sensing data. Although we carefully checked them and removed some obvious problematic samples, there may still be some uncertainties caused by the inconsistency. To overcome this problem, the planned sampling section can be divided into stable and unstable illumination zones. For stable zones, the lighting is steady, and therefore, the illuminance cannot be measured strictly synchronously with the satellite overpass. For unstable zones, the lighting is changeable; therefore, the illuminance should be recorded strictly synchronously with the satellite overpass. Assigning more observation teams also helps to reduce the inconsistency because they can collect abundant samples in less time. Moreover, there are also a lot of pixels that were affected by these factors in the remotely sensed illuminance map. How to identify and remove these problematic pixels is still a difficult problem to overcome in the future. During this in situ observation, normal handheld GNSS instruments were used to record sample coordinates. However, the positioning accuracy of these instruments is generally within 10 m. Given the obvious spatial difference in the light environment, this accuracy level is not sufficient. RTK GNSS instruments with centimeter-level accuracy can effectively improve the spatial consistency between field observations and remote sensing data. There is another important point to note, which is the observation angle of the satellite. Though JL1-3B can image within the off-nadir angle of $\pm 45°$, the satellite zenith angle should not be high enough to eliminate the shading effect of buildings and trees, and also reduce the influence of Rayleigh scattering of the atmosphere.

## 6. Conclusions

This study proposed a new space-borne method for evaluating road lighting quality based on JL1-3B nighttime light remote sensing data. Firstly, synchronous field observations were carried out to measure illuminance in Nanjing, China. After a series of preprocessing of the JL1-3B image, the in situ observed illuminance and the radiance of JL1-3B were combined to map the high-resolution surface illuminance based on the close relationship between them. Two models (multiple linear regression and random forest) with six independent variable combinations were employed and compared to develop the optimal model for illuminance estimation. Results showed that the random forest model with Hue, Saturability, lnI, lnR, lnG and lnB as the independent variables achieved the best performance ($R^2$ = 0.75, RMSE = 9.79 lux). Additionally, the optimal model was applied to the JL1-3B preprocessed image to derive the surface illuminance map. The average, standard deviation and $C_v$ of the illuminance within roads were calculated to assess their lighting quality.

This study is a preliminary study to develop a technical framework to evaluate road lighting quality using JL1-3B nighttime light remote sensing data. JL1-3B can be ordered from Changguang Co., Ltd., Changchun, China and the instruments for ground-based observation were inexpensive TES-1399R illuminance meters. The devices and methods used in this study have the advantages of low cost, simplicity and reliability, providing a reference for road lighting quality evolution in other regions.

**Supplementary Materials:** The following supporting information can be downloaded at: https://www.mdpi.com/article/10.3390/rs14184497/s1, Figure S1: The boxplot of the extracted roads; Table S1: Road lighting quality evaluation metrics of the extracted roads.

**Author Contributions:** Conceptualization, Y.X.; methodology, Y.X. and N.X.; formal analysis, N.X., Y.X. and B.W.; investigation, Y.X., Y.Y., X.Z., Z.G. and B.W.; writing—original draft preparation, N.X. and Y.X.; writing—review and editing, N.X. and Y.X.; supervision, Y.X.; project administration, Y.X. All authors have read and agreed to the published version of the manuscript.

**Funding:** This research was funded by the National Natural Science Foundation of China (41871028, 42171101) and the Science and Technology Research Plan in Key areas of Xinjiang Production and Construction Corps (2022AB016).

**Conflicts of Interest:** The authors declare no conflict of interest.

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
