# Peer review of "Evaluating Road Lighting Quality Using High-Resolution JL1-3B Nighttime Light Remote Sensing Data: A Case Study in Nanjing, China"

_remotesensing, doi:10.3390/rs14184497_

Round 1
Reviewer 1 Report
This paper explored the potential of JL1-3B nighttime light remote sensing images to evaluate road lighting quality. Random forest method with six independent variable combinations were employed to develop the optimal model for illuminance estimation. I found this topic interesting and would provide valuable information. There are a few points that could be further improved. Please see my comments below.
1. The literature review needs to include the analysis of current studies on road light extraction or evaluation using nighttime light data, especially using JL1-3B data. For example, reference 26 used JL1-3B data to extract and classify street lights.
2. Figure 6. I suggest adding fitted line to the plot to better illustrate the relationship between the observed illuminance and the radiance of three JL1-3B bands. Actually, the correlation is relatively good from the correlation coefficients, but it is not very obvious from the scatter plots.
3. Page 8 Section 4.1. Could the authors add more analyses to illustrate the strengths and weaknesses of this approach? For example, the analysis of residual distribution can be used to explain under what circumstances the method can accurately estimate the illuminance and under what circumstances it cannot.
4. Page 10 Section 4.2. Could the authors explain why only 7 typical roads were selected for analysis instead of all roads? The number of 7 is relatively small and may not be sufficient to demonstrate the road lighting quality of the entire study area.
Reviewer 2 Report
Accurate and timely knowledge of road lighting illuminance is meaningful for the planning and management of urban road lighting systems. The author explored the road lighting mapping with JL1-3B and evaluated its accuracy, which is of great significance to enrich the cognition of road lighting quality, but there are still several problems to be solved.
1. The evaluation of road lighting quality in this paper is limited to the scale of 7 typical roads, and the resolution advantage of L1-3B data is not reflected. Add the evaluation of road lighting quality from the scale of pixel.
2. Discussion and conclusion need to be rewritten. The first and second paragraphs of discussion belong to the contents of conclusion and need to be simplified and integrated into the conclusion; The discussion needs to be deepened to explain the problems and reasons of the L1-3B data in the surface illuminance estimation.
3. 2.2.2. In field measurements, complete the information of field sampling data. How many samples are collected, how many samples are used for model training and how many are used for model verification.
4. Explain the relationship between road lighting quality and illuminance at the beginning of the paper, to facilitate readers' understanding.
5. In figure 3 and section 3.4, add the data source of actual road lighting.
6. Move the relevant words of line286 go 288 to 3.4 to reflect the judgment standard of road lighting quality.
7. Add the estimation accuracy of road spatial scopes.
8. Line 292, "the other roads qualified the standard" is not rigorous enough. The average value of Huayuan road has exceeded the standard range. Should it have quality problems?
9. Line 312 to 313, “Fujian Road and Guandong road needed to improve the uniformity”, gives the basis.
10. In Table 5, add the national standard values. In addition, explain why select these seven roads to evaluate the light quality.
Reviewer 3 Report
I think the manuscript is well written and should be accepted for publication after minor revision.
My concern on the research design and the conclusion is the use of only single scene of satellite with one hour of ground data collection. The brightness observed by satellites are influenced by various factors such as cloud condition, the presence of several natural light sources especially the moon etc.
The words and sentences that should be considered for revision are listed below.
(1) L73-74
Satellite based nighttime light data including JL1-3B is also limited by weather, especially the cloud.
(2) L76
Information such as how much does a scene of JL1-3B cost should be included to backup the claim of the feasibility proposed methodology at larger geographic scales.
(3) Equation (1)
What is h in 'h - 360'?
(4) L157-158
'i' in 'ai' and 'bi' should be subscripts.
(5) L173-174
Does the problematic sample points mean the points with other ground based light sources such as vehicles' headlights or nearby buildings? Since the purpose is to identify insufficient road lighting areas with satellite images, I am not sure if such exclusion makes sense.
(6) Table 2
Is the six combinations of independent variables subjectively picked? Please provide theoretical justification if any?
(7) L247-249
'was more appropriate for'. Do the authors mean 'was more appropriate than'?
Round 2
Reviewer 1 Report
Thanks to the authors' revision. The authors added in-deep analysis and have fully addressed my comments. I think this paper can be accepted in preset form.